# Performance of PVDF-TiO_2_ Membranes during Photo-Filtration in the Presence of Inorganic and Organic Components

**DOI:** 10.3390/membranes12020245

**Published:** 2022-02-21

**Authors:** Duc-Trung Tran, Julie Mendret, Jean-Pierre Méricq, Catherine Faur, Stephan Brosillon

**Affiliations:** IEM (Institut Européen des Membranes), UMR 5635 (CNRS-ENSCM-UM2), Université de Montpellier, 34095 Montpellier, France; duc-trung.tran@umontpellier.fr (D.-T.T.); jean-pierre.mericq@umontpellier.fr (J.-P.M.); catherine.faur@umontpellier.fr (C.F.); stephan.brosillon@umontpellier.fr (S.B.)

**Keywords:** inorganic fouling, organic fouling, photo-filtration, PVDF-TiO_2_ membrane

## Abstract

In this study, the anti-fouling performance of PVDF-TiO_2_ composite membranes, indicated by their permeate flux, was studied with different types of synthetic feed solutions. Photo-filtration (filtration under continuous UV irradiation) of solutions containing inorganic and organic components, which are ubiquitous in drinking/natural water, was performed to evaluate their influence on the photo-induced properties and performance of the membranes. The results indicated that inorganic fouling was unlikely to occur on PVDF-TiO_2_ membranes, and the presence of common inorganic ions in drinking water did not hinder their performance. However, in the particular case where a small amount of Cu^2+^ coexisted alongside HCO_3_^−^ in the feed solution, inorganic fouling occurred, causing severe flux decline and prohibiting the photo-induced properties of the membranes. On the other hand, when used to filter organic fouling solutions, the membranes showed strong resistance to sodium alginate fouling, and less so for humic acids. In terms of separation efficiency, the membranes showed no advantages when operated in photo-filtration mode, as the rejection rate of both foulants under photo-filtration was not higher than that under normal filtration. In the case of humic acids, the photodegradation of humic substances into smaller compounds that were able to enter the permeate stream led to a lower rejection rate. Nevertheless, photo-filtration of these organic foulants still offered a higher permeate flux than normal filtration, up to a certain concentration level (5 mg/L for humic acids and 50 mg/L for sodium alginate).

## 1. Introduction

In the field of water treatment, advanced oxidation processes (AOPs) are often used during the tertiary step to achieve high quality of water. With AOPs, pollutants are degraded into simple and harmless molecules without generating secondary waste, thanks to the reactions of one of the strongest oxidants in aqueous solution, i.e., the hydroxyl radical [1]. The generation of such hydroxyl radicals can be from a primary oxidant, an energy source, or a catalyst. Among those methods, TiO_2_ photocatalyst has attracted strong interest from researchers for half a century due to its excellent photocatalytic activities [2] and photo-induced superhydrophilicity [3]. On the other hand, membrane technology such as ultrafiltration (UF)—another advanced process for water treatment—offers high efficiency in the removal of particles, microorganisms, or some dissolved compounds by the principle of physical separation. However, membrane processes are still limited by membrane fouling. Thus, the idea of combining AOPs and membrane separation, specifically TiO_2_ and membrane, into a single treatment unit, usually called “photocatalytic membrane reactor” (PMR), has been explored recently in order to overcome membrane processes limitations [4,5]. A PMR can either be configured with suspended catalysts or catalysts immobilized in/on the membrane, of which the latter configuration is often preferred for process intensification because it can combine chemical reactions and separation in a single step while being able to limit photocatalyst leaching [6].

As a result, several research groups have successfully prepared composite polymer-TiO_2_ membranes via a large range of methods, including surface modification; sol–gel; interfacial polymerization; and, most popularly, phase separation [7]. The method typically involves the coating of TiO_2_ nanoparticles on the surface of a ready-made membrane, or blending the TiO_2_ nanoparticles into the polymer solution that subsequently will be used to fabricate the membrane. Thanks to the hydrophilic nature of TiO_2_, the composite membranes showed improved permeability as well as better anti-fouling performance compared to neat polymeric membranes [8,9]. In addition, the photocatalytic activities of TiO_2_ under UV light have also been utilized to induce self-cleaning, anti-bacterial functionalities to the composite membranes [10,11,12]. Researchers also attempted to investigate the structure-influencing parameters during membrane preparation to optimize its performance by modifying the base polymer material [13,14], varying the concentration of casting solution components (most importantly the TiO_2_ concentration) [10,15], or introducing additives [16,17].

Nevertheless, even if the membrane can be synthesized with the best preparation parameters, its performance is still under the influence of, and sometimes restricted by, other filtration conditions. In a previous study, we investigated how UV irradiation methods can affect the membrane performance in terms of its pure water flux [18]. Moreover, the photocatalytic activity of TiO_2_ was proven to be less effective in certain aqueous media. For example, trace quantities of transition metal ions such as Cu^2+^ and Fe^3+^ could suppress the photo-degradation of dyes in TiO_2_ suspensions under visible irradiation [19]. Evidences of an inorganic salt layer on the TiO_2_ surface, which inhibited the adsorption of dyes or decreased the rate of photocatalytic degradation under UV, were also found [20,21]. In addition, inorganic anions, especially carbonate and phosphate, might also interact with TiO_2_ and influence the bulk concentration of hydroxyl radical, thus inhibiting the reactivity of TiO_2_ nanoparticles [22]. Since such species are commonly found in most types of water, the performance efficiency of composite polymer-TiO_2_ membranes is subject to their presence in the feed.

To our knowledge, despite the significant number of studies on polymer-TiO_2_ membranes, the focus was mostly on anti-fouling performance or photocatalytic degradation of organic pollutants. No attempt has been made to study membrane performance under the influence of inorganic species, which are commonly present in water. Thus, in this study, we aimed first to contribute more insights into the performance of TiO_2_-based composite membranes under the influence of inorganic species. Moreover, another aspect of PVDF-TiO_2_ membranes is their anti-fouling performance when operated under continuous UV irradiation. Since other studies usually focus on either the intrinsic properties or the cleaning properties of photocatalytic membranes, less attention was paid in terms of the membrane photo-filtration performance, which was demonstrated as a very promising method to increase the permeate flux [18], with organic foulants.

As a result, the membrane preparation and characterization aspects are only briefly mentioned in this study. Instead, the focus is on the photo-filtration performance of PVDF-TiO_2_ membranes with different types of water. Filtration of solutions containing common inorganic ions and organic foulants, which are representative in drinking water and natural water, respectively, was performed with and without UV irradiation. The permeate flux, as well as the separation rate under UV for organic foulants, were monitored as an indication of photo-filtration performance, while the membrane surface after filtration tests was analyzed with different tools to study the fouling behavior.

## 2. Materials and Method

### 2.1. Materials

PVDF pellets (Sigma Aldrich, Saint Louis, MO, USA) with a molecular weight (Mw) of 275 kDa were used as the polymer material. The solvent was N-N-dimethylacetamide (DMAc, purity > 99.5%, Sigma-Aldrich) while the additive was polyethylene glycol (PEG200, average Mw 200 Da, Sigma-Aldrich). The nanoparticles were Aeroxide TiO_2_ P25 nanopowder (approximately 85% anatase and 15% rutile, size ≈ 21 nm, purity > 99.5%, Sigma-Aldrich). For filtration tests with inorganic components, the following salts were used to provide the ions in the feed stream: NaHCO_3_, NaCl, NaNO_3_, K_2_SO_4_, K_2_HPO_4_, CuSO_4_, and ZnSO_4_. For filtration tests with organic components, humic acid (HA) and sodium alginate (SA) were used as the model foulants. All chemicals have purity >99%, were purchased from Sigma-Aldrich and used as received.

### 2.2. Membrane Preparation

The casting solutions were prepared by mixing PVDF (20 wt %), TiO_2_ (20 wt % TiO_2_/PVDF, i.e., the TiO_2_ concentration is expressed relatively to PVDF), and PEG200 (5 wt %) in DMAc. The mixture was sonicated for 20 min and then agitated for 24 h at 50 °C by a magnetic stirrer to obtain a homogeneous solution. A Teflon sheet was taped on top of a glass plate to form the casting support, wherein the solution was cast upon using an automated casting knife (Erichsen, Hemer, Germany) set at 250 μm thickness and 50 mm·s^−1^ velocity. The temperature of the dope solution at the time of casting was 50 °C, while the temperature at the surface of the Teflon sheet was also maintained at 50 °C by means of a heating plate and verified by an infrared thermometer (Testo 845, Testo, West Chester, PA, USA). After casting, the plate was immediately immersed in a 50 °C deionized water bath (the exposure time of the cast film to ambient air before immersion was less than 10 s). The temperature of 50 °C was chosen throughout the procedure to ensure membrane rigidity, which was a result obtained previously [23]. The plate was left in the coagulation bath for 12 h, and then the prepared membrane was detached from the Teflon support, rinsed thoroughly with deionized water to remove the remaining solvent, and stored in Milli-Q^®^ water (18 MΩ.cm resistivity) at room temperature (20 ± 1 °C) under dark conditions.

### 2.3. Photo-Filtration Procedure

#### 2.3.1. Apparatus

Photo-filtration tests were performed in a specifically designed crossflow filtration cell built with a Quartz window on top so that the membrane could be illuminated by a UV lamp (Philips PL-S 9W, λmax = 365 nm). The membrane active filtration area was 30 cm^2^, while the irradiated area was considered to be equal to the filtration area. The UV irradiance at the membrane surface was validated by a radiometer (UVA-365, Lutron, Coopersburg, PA, USA). Transmembrane pressure (TMP) was provided by compressed air and controlled in the range of 0–1.5 bar. The feed solution was placed in a 5 L compressed tank and circulated at a cross-flow velocity of around 0.1 m·s^−1^ using a peristaltic pump. Permeate flux was determined by monitoring the permeate mass via an electronic balance (Ohaus, Parsippany, NJ, USA, precision of 0.01 g) and recorded automatically by a computer software. Prior to each filtration test, the membrane was compacted by pure water filtration for 90 min, in which the pressure was increased stepwise from 0.25 to 0.5, 0.75, and 1 bar every 15 min, and finally to 1.25 bar for 30 min. All experiments were performed at room temperature (20 ± 1 °C), while the permeate temperature was monitored and all permeate fluxes were subsequently corrected to their corresponding values at 20 °C.

#### 2.3.2. Filtration of Inorganic Contents

Photo-filtration tests of inorganic solutions started without UV irradiation for the first 30 min, followed by 60 min of filtration with UV at irradiance I = 1 mW/cm^2^, then ended with another 30 min of filtration without UV. Such irradiance (1 mW/cm^2^) was proven to be sufficient for photo-induced activities of TiO_2_ in our previous study [18]. The transmembrane pressure was maintained constantly at 1 bar.

Filtration tests were first performed with tap water (collected in Montpellier, France) and Hépar^®^ mineral water (Nestle France, Issy-les-Moulineaux, France). Both types of water contained a negligible amount of organics (total organic carbon ≤ 1 mg/L) but were rich in inorganic contents. Their ionic composition is presented in Table 1 along with that of Milli-Q water as a reference. Afterwards, synthetic solutions containing specific inorganic ions of known concentrations, which are explained in Section 3.1, were used to study the effects of each ion. It should be noted that for each test involving one or two targeted ions, their counter anions/cations in the salts were also present in the solutions.

#### 2.3.3. Filtration of Organic Contents

Photo-filtration tests of organic species were performed with typical model foulants, namely, humic acids (HA) and sodium alginate (SA). Fresh solutions of SA (5–50 mg/L) were prepared by dissolving the appropriate amounts of SA in Milli-Q^®^ water, while HA solutions (1.25–10 mg/L) were prepared by dissolving HA in 0.01 M NaOH solution. The concentrations of HA and SA were chosen to reflect the fractions of humic substances and extracellular polymeric substances, respectively, in the natural organic matters present in surface water [24,25,26].

For photo-filtration of organic foulants, UV irradiation was maintained for the whole duration of the tests, while separate tests without UV were also performed as references. Importantly, the operating pressure was reduced to 0.5 bar so that membranes suffered virtually no compaction, and thus any flux decline could only be attributed to be the result of organic fouling.

### 2.4. Analytical Tools

The inorganic composition of major ionic elements in water was analyzed by ionic chromatography (Dionex ICS-1000 system, Thermo Scientific, Waltham, MA, USA) and titration method (Rondolino G20, Mettler Toledo, Columbus, OH, USA). The trace analysis of metal elements in water was analyzed by inductively coupled plasma mass spectrometry (ICP-MS) (ICAP Q, Thermo Scientific). Concentration of HA was determined by measuring the HA solution absorbance at 254 nm using a UV–VIS spectrophotometer, while concentration of SA was determined by total organic carbon (TOC) measurement using a TOC analyzer (TOC-V_CSH_, Shimadzu, Kyoto, Japan).

The membrane surfaces after filtration tests were examined by energy dispersive X-ray (EDX) spectroscopy (EVO HD, Thermo Scientific) and Fourier transform infrared (FTIR) spectroscopy (Nicolet Nexus 710, Thermo Scientific).

## 3. Results and Discussion

The properties of PVDF-TiO_2_ membranes prepared in this study were thoroughly characterized in our previous studies [18,23]. In brief, the membranes had an asymmetric structure typically found in membranes prepared by phase separation, with a thin top layer with visible open pores on the surface and a porous cross section with some finger-like macrovoids. The maximum pore diameter on the surface was around 70 nm. The membranes possessed a permeability at 20 °C of around 125 L·h^−1^·m^−2^.bar^−1^ and a porosity of around 75%. The distribution of TiO_2_ across the membranes was very uniform, both on the surface and on the cross-section. PVDF-TiO_2_ membranes in general had superior properties compared to their neat PVDF counterpart, yet their most interesting features lied in the photo-induced properties. Thus, in this study, the focus is only on their photo-filtration performance, i.e., their filtration performance under continuous UV irradiation.

### 3.1. Influence of Inorganic Contents in Drinking Water on Photo-Filtration Performance

#### 3.1.1. Photo-Filtration of Drinking Water

As demonstrated previously, when photo-filtration was performed with pure water, PVDF-TiO_2_ membranes became more hydrophilic thanks to the photo-induced hydrophilicity effect of TiO_2_, which led to an increase in permeate flux [18]. As shown in Figure 1a, the same phenomenon was also observed for the mineral water Hepar^®^, which contained a mixture of common ions in drinking water (Table 1). After the natural flux decline due to membrane compaction during the first 30 min of filtration without UV, the permeate flux gradually increased upon UV irradiation. This suggests that these ions, at least at the concentrations reported (Table 1), did not affect the efficiency of photo-filtration. In other words, mineral scaling was unlikely to occur to inhibit the interactions between TiO_2_ nanoparticles and UV radiations.

However, when the same experiment was performed with tap water, the flux rising period could no longer be observed, as permeate flux kept decreasing for the whole test, regardless of UV irradiation (Figure 1b). In addition, the top surface of the membrane submitted to UV showed a light brownish color, possibly as a result of an inorganic fouling layer. The fact that this brownish layer did not appear on the half-membrane surface covered from UV (inset of Figure 1b) suggests that this layer was formed as a result of some photocatalytic reactions. For comparison, the surface of the membrane photo-filtrated with mineral water showed no irregularities when compared with the membrane surface half-covered from UV (inset of Figure 1a). Since the concentration of most ions in the mineral water was significantly higher than that in the tap water (Table 1), it can be assumed that these ions were not directly responsible for the fouling behavior in photo-filtration of tap water. Since the organic content in tap water was insignificant as well, it can be predicted that some inorganic trace elements present in the tap water but not in the mineral one might be responsible for this phenomenon, as inactivation of TiO_2_ photocatalysis was previously reported to occur with the presence of some transition metals in water [19]. Thus, comprehensive trace analysis of most metal elements was performed, and the results revealed that only in the tap water existed some small amounts of Cu and Zn (Table 1), which was probably the result of contamination during water supply. It is thus assumed that the interactions between either of these two metals and the common ions in tap water prohibited the photo-induced properties of PVDF-TiO_2_ membranes and also led to such fouling effect.

#### 3.1.2. Influence of Metal Cations on Photo-Filtration Efficiency

To study the effect of metal elements in conjunction with common inorganic ions on photo-filtration performance of PVDF-TiO_2_ membranes, we dissolved small amounts of Cu or Zn salts (CuSO_4_ or ZnSO_4_) in the Hepar^®^ mineral water to provide the feed solutions, as Cu and Zn were identified as the possible metallic contaminants in tap water. The concentrations of Cu^2+^ and Zn^2+^ were 5 and 2.5 mg/L, which was about 10 and 15 times higher than that found in the tap water (Table 1), respectively, in order to accentuate any potential effects. Figure 2 shows the flux behavior during photo-filtration of these solutions. The flux behavior of the Zn-doped mineral water (Figure 2a) was similar to that in the mineral water photo-filtration test, and the membrane surface after the test appeared normally (inset of Figure 2a), which indicates that the presence of Zn^2+^ in drinking water did not interfere with the photo-induced properties of PVDF-TiO_2_ membranes or cause inorganic fouling. On the other hand, it was quite apparent that the ion responsible for inorganic fouling in tap water filtration was Cu^2+^, as a constant flux decrease was observed during photo-filtration of the Cu-doped mineral water (Figure 2b), and more evidently, the irradiated membrane surface after the test showed a light brownish color (inset of Figure 2b) similar to that found in the tap water photo-filtration test (Figure 1b). Analysis of this membrane surface by EDX revealed the deposition of scaling clusters rich in Cu content (42.5 wt %), as can be seen in Figure 3.

#### 3.1.3. Influence of Anions on Photo-Filtration Efficiency

Although Cu^2+^ was identified to affect the performance of PVDF-TiO_2_ membranes during photo-filtration, it is unlikely to act as an isolated ion: the inorganic fouling layer could be the result of some combinations between Cu^2+^ and other ions present in the drinking water. To understand this inorganic fouling mechanism, we performed separate photo-filtration tests in which Cu^2+^ was paired in pure water with other common anions (HCO_3_^−^, HPO_4_^2^, NO_3_^−^, Cl^−^, SO_4_^2−^). The concentration of Cu^2+^ was similar to that found in the tap water (0.5 mg/L), and the concentrations of the anions were modeled on the basis of those in the mineral water (Table 1), as the concentration of most anions in the mineral water was higher than that in the tap water. The permeate flux behavior of these tests is reported in Figure 4. It can be seen that the combinations of ions such as HPO_4_^2−^, NO_3_^−^, Cl^−^, and SO_4_^2−^ with Cu^2+^ did not lead to any irregularities in flux behavior during UV irradiation, suggesting both mineral scaling and the photo-inhibitory effect did not occur. It should also be noted that the presence of the counter anion of Cu^2+^, which was SO_4_^2−^, and the counter cations of the anions, which were Na^+^ and K^+^, had no effect on the process either, as validated by separated single tests. However, the presence of both Cu^2+^ and HCO_3_^−^ in the feed solution led to the flux inhibitory effect already observed in tap water photo-filtration. In addition, the brownish deposition on the membrane surface was only observed for this test as well. Similar to the case with Cu-doped mineral water, analysis via EDX mapping showed an abundant distribution of Cu clusters on the membrane surface after the photo-filtration test with Cu^2+^ and HCO_3_^−^, as can be seen in Figure 5. Since this phenomenon did not occur with other anions, it can be concluded that the interactions between Cu^2+^ and HCO_3_^−^ during the photo-filtration test led to the reduced performance of PVDF-TiO_2_ membranes.

#### 3.1.4. Discussion

To further understand the negative influence of inorganic ions (specifically Cu^2+^ and HCO_3_^−^) on the performance of PVDF-TiO_2_ membranes, we performed the filtration of feed solutions containing 0.4 mg/L Cu^2+^ and 400 mg/L HCO_3_^−^ (plus 0.6 mg/L SO_4_^2−^ and 150 mg/L Na^+^ as counter ions) under various conditions of operation (without UV; without oxygen). The filtration performance of PVDF-TiO_2_ membrane towards the solution containing Cu^2+^ and HCO_3_^−^ was also compared with that of a neat PVDF membrane. The results are presented in Figure 6.

As discussed previously, the PVDF-TiO_2_ membrane surface after photo-filtration of a Cu^2+^/HCO_3_^−^ solution showed a light brownish color (Figure 6a). Yet, when the same filtration test was performed without UV irradiation, although the flux expectedly saw no increase due to the lack of photo-induced hydrophilicity (Figure 6b), the membrane surface only showed a light blue color, which was possibly the precipitation of copper carbonate salt. This means that the brownish color on the membrane surface after photo-filtration was the result of photocatalytic reactions induced under UV irradiation only. To verify this hypothesis, we performed photo-filtration tests under deoxygenated condition and with neat PVDF membrane, as free oxygen molecules play a role in determining the efficiency of TiO_2_-induced photocatalytic reactions. The deoxygenated photo-filtration test was performed by purging the feed solution with N_2_ for 90 min prior to filtration, while the transmembrane pressure was provided in the form of compressed N_2_ instead of compressed air. The result, under UV irradiation, showed a similar flux behavior to that observed in the presence of oxygen, but it seemed that the photo-reactions were stronger without oxygen in the system, suggested by a more intense brownish color on the membrane surface (Figure 6c). Mass analysis from EDX was in agreement with this observation, as the mass ratio of Cu on the membrane surface was 1.84±0.09% for the photo-filtration and 3.46±0.07% for the deoxygenated photo-filtration test. As oxygen may function as the electron trap during photocatalytic reactions, it is possible that without oxygen, any potential reactions involving the ions (Cu^2+^ and HCO_3_^−^) could be further enhanced, which would lead to a severe deposition on the membrane surface. In addition, photo-filtration test using the neat PVDF membrane validated the fact that without the photocatalyst TiO_2_, UV alone was not responsible for such photo-reactions on the membrane surface, as suggested by the unchanged color of the membrane after the test (Figure 6d). 

From these results, the following hypotheses can be proposed for the phenomenon that occurred during photo-filtration of solutions containing Cu^2+^ and HCO_3_^−^: (i) the precipitation of the copper carbonate salt leads to inorganic fouling on the membrane, decreasing its permeate flux; (ii) the formation and subsequent deposition on the membrane of this precipitate block TiO_2_ nanoparticles from interactions with UV radiation, and thus photo-induced hydrophilicity cannot occur in order to induce the flux increase during UV irradiation; (iii) possible photo-reactions involving Cu^2+^, HCO_3_^−^, TiO_2_, and UV radiations lead to the formation of a Cu-based complex (with brownish color) that deposits on the membrane and further decrease its filtration performance. 

As a general conclusion for this part, it can be said that the photo-filtration performance of PVDF-TiO_2_ membranes will not be affected by the inorganic ions in typical drinking water, which is the expected feature of micro- and ultra-filtration processes. The only exception found is when a small amount of Cu^2+^ is present in water in concurrent with the hydrocarbonate ion. Since the latter is ubiquitous in most types of water, the prevention of the former ion, which can come as a contaminant from water transport by piping, is thus very important to maintain the efficiency of photo-filtration

### 3.2. Influence of Organic Contents in Natural Water on Photo-Filtration Performance

#### 3.2.1. Humic Acids

Humic acids (HA) are representative of natural organic matter (NOM) in surface water and have been used extensively as the model foulant in MF and UF studies. Since the UV-blocking effect can occur very quickly with organic foulants, photo-filtration tests with organic contents were performed on a full-time irradiation basis, and the results were compared with those of normal filtration tests without UV for HA concentrations ranging from 1.25 to 10 mg/L (Figure 7).

It appears that for HA, improved photo-filtration performance in terms of permeate flux could only be obtained at concentrations less than 5 mg/L. There were clear distinctions in flux trend when the HA concentration was 1.25 mg/L, wherein the flux with UV showed a gradual increase similar to the case of photo-filtration with pure water, or at 2.5 mg/L, wherein the flux remained stable compared to the continual flux decrease during filtration without UV. However, at HA concentrations of 5 mg/L and above, there was no difference in flux behavior whether UV was involved or not. In fact, even when the irradiance was increased to 2.5 mW.cm^−2^ (the highest possible intensity for this apparatus), the flux behavior was almost the same with that at 1 mW.cm^−2^ (Figure 7d). The concentration of HA thus obviously plays an important role in the photo-filtration performance of PVDF-TiO_2_ membranes. At a critical level, the degree of fouling became so severe that TiO_2_ nanoparticles can no longer be exposed effectively to UV irradiation. Thus, the photo-induced hydrophilicity effect could not occur, and the flux continued to decrease as a result of organic fouling. Nevertheless, the membrane still demonstrated its superior performance in photo-filtration mode if HA concentration can be controlled under a certain level.

On the other hand, photo-filtration appeared to cause a negative effect in terms of separation rate, as can be seen in Figure 8. At all concentrations, the rejection rate for photo-filtration was always lower than that for filtration without UV: for a HA concentration of 5 mg/L, the maximum rejection under UV was only 64% compared to 81% without UV. In return, the rejection rate did not depend on the UV irradiance value, as shown by similar rejection rates of a 10 mg/L HA solution under 1 and 2.5 mW/cm^2^. One possible explanation could be that the photocatalytic reactions taking place on the membrane surface to degrade HA would create some by-products with lower molecular weights. These lower Mw molecules would be able to pass through the membrane pores, increasing the content of HA-related compounds in the permeate and thus decreasing the rejection. Such formations of lower molecular size and higher UV absorbing compounds were also reported in studies on photocatalytic degradation of HA [27,28]. In addition, FTIR spectroscopy analysis of the membrane surface before and after filtration, with and without UV irradiation (Figure 9), also supports this hypothesis. In Figure 9a, the peaks at 1403, 1181, 1071, 876, 839, and 762 cm^−1^ are characteristic peaks for the different crystalline phases of PVDF [29]. More importantly, characteristic peaks of HA were identified at 1101, 1030, and 1009 cm^−1^ (Figure 9b), indicating a sorption mechanism of HA on the membrane. Furthermore, the intensity of these peaks increased dramatically for the irradiated membrane (Figure 9c), suggesting significant changes in chemical composition of the compounds deposited on the membrane surface as a consequence of photocatalytic degradation of HA.

#### 3.2.2. Sodium Alginate

Sodium alginate represents the fraction of EPS present in natural water and has been known as a model organic foulant in MF and UF. Results from photo-filtration tests of SA (Figure 10) suggested that the PVDF-TiO_2_ membranes have a higher tolerance for SA fouling compared to HA.

As can be seen in Figure 10, photo-filtration of SA at 10 mg/L showed strong improvement in terms of permeate flux compared to the case of filtration without UV. Although the flux enhancement was less visible for photo-filtration of SA at 50 mg/L, the permeate flux behavior at such high concentration of foulant demonstrated that PVDF-TiO_2_ membrane was very effective for SA anti-fouling performance. In addition, for SA, the rejection rate of filtration tests with and without UV were almost the same at all feed concentrations (Figure 11), while the FTIR spectra of the pristine membrane, membrane after filtration without UV, and membrane after photo-filtration of SA were almost identical, with only characteristic peaks for PVDF identified (Figure 12). These results suggest that the SA molecules were not sorbed on the membrane, even when the photo-induced properties of the membranes were not activated. In fact, the permeate flux in filtration tests of SA without UV showed little decrease over time (Figure 10), suggesting the excellent resistance of PVDF-TiO_2_ membranes to SA fouling. On the other hand, SA seemed to not be affected by photocatalytic activities, and thus the photo-filtration of solutions containing SA was only beneficial in terms of improving the permeate flux thanks to the photo-induced hydrophilicity effect.

Conclusively, it can be said that the performance of PVDF-TiO_2_ membranes can be improved when operated in photo-filtration mode for filtration of water containing organic contents, except for a drawback in rejection in the case the foulant is humic acid. Obviously above a certain concentration of the foulants, the enhancement in permeate flux also diminishes due to the stronger rate of fouling, yet no membranes can sustain at every condition and thus the advantages of PVDF-TiO_2_ photocatalytic membranes are still demonstrated.

## 4. Conclusions

In this study, the (photo-)filtration performance of PVDF-TiO_2_ membrane was examined with different types of feed solution: inorganic-based and organic-based. Results from photo-filtration of solutions with inorganic contents demonstrated that inorganic fouling did not occur in the presence of common ions in drinking water (HCO_3_^−^, Cl^−^, NO_3_^−^, SO_4_^2−^, Na^+^, K^+^, Mg^2+^, Ca^2+^). However, if a very small amount (approximately 0.5 mg/L) of Cu^2+^ exists in the solution, it can combine with HCO_3_^−^ to cause inorganic fouling, which severely decreases the membrane permeate flux and limits its photo-induced properties. Potential photo-reactions also occurred under photo-filtration condition to form a brownish copper-based complex that was deposited on the membrane surface. Further studies are required to identify the mechanism behind this phenomenon.

Photo-filtration tests of HA and SA as representative organic foulants in natural water both showed improvements in terms of permeate flux up to a certain foulant concentration (5 mg/L for HA and 50 mg/L for SA). The membranes showed strong resistance to SA fouling as the SA molecules showed little tendency to be adsorbed on the membrane surface even when filtration was performed without UV. The rejection rate of SA did not improve with photo-filtration as well. On the other hand, HA was not only adsorbed on the membrane but also degraded under photocatalytic activities of the membranes, probably leading to the formation of molecules with lower molecular weights that can pass through the membrane pores. Thus, the rejection rate of HA was actually lower when photo-filtration was used compared with filtration without UV. In conclusion, the photo-filtration method shows effectiveness for organic solution filtration in terms of increasing the permeate flux and can be envisaged as a solution to mitigate organic fouling. Lastly, further investigations into the effectiveness of PVDF-TiO_2_ membrane photo-filtration, for example, filtration of both inorganic and organic contents, a mixture of foulants in the feed, or the feasibility of using irradiation cycles for fouling solutions filtration, are advised in order to obtain a better understanding of the membrane behavior under different circumstances.

## Figures and Tables

**Figure 1 membranes-12-00245-f001:**
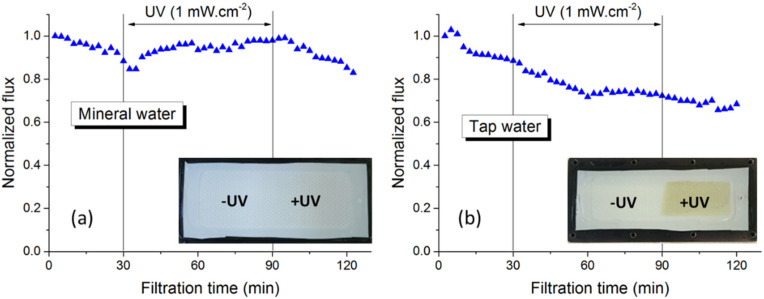
Flux behavior during photo-filtration of (**a**) mineral water and (**b**) tap water.

**Figure 2 membranes-12-00245-f002:**
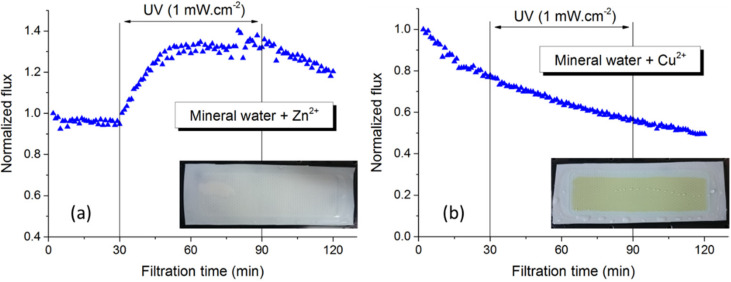
Flux behavior during photo-filtration of mineral water doped with (**a**) Zn and (**b**) Cu.

**Figure 3 membranes-12-00245-f003:**
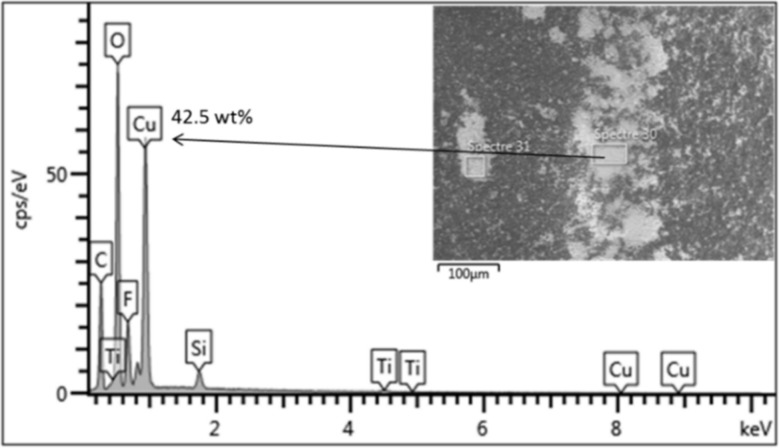
EDX analysis of membrane surface after photo-filtration with Cu-doped mineral water.

**Figure 4 membranes-12-00245-f004:**
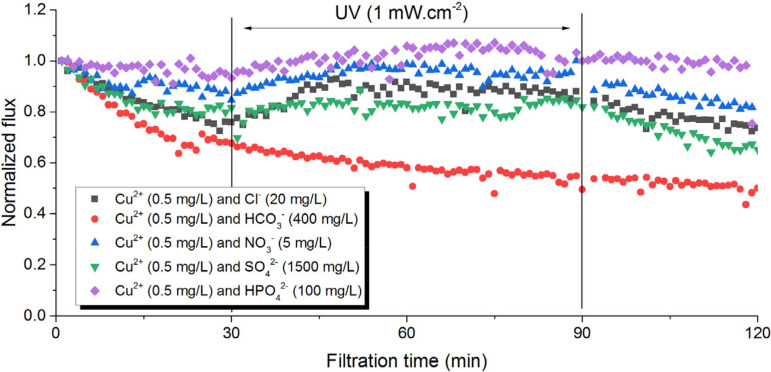
Flux behavior during photo-filtration of solutions containing Cu^2+^ and other inorganic ions.

**Figure 5 membranes-12-00245-f005:**
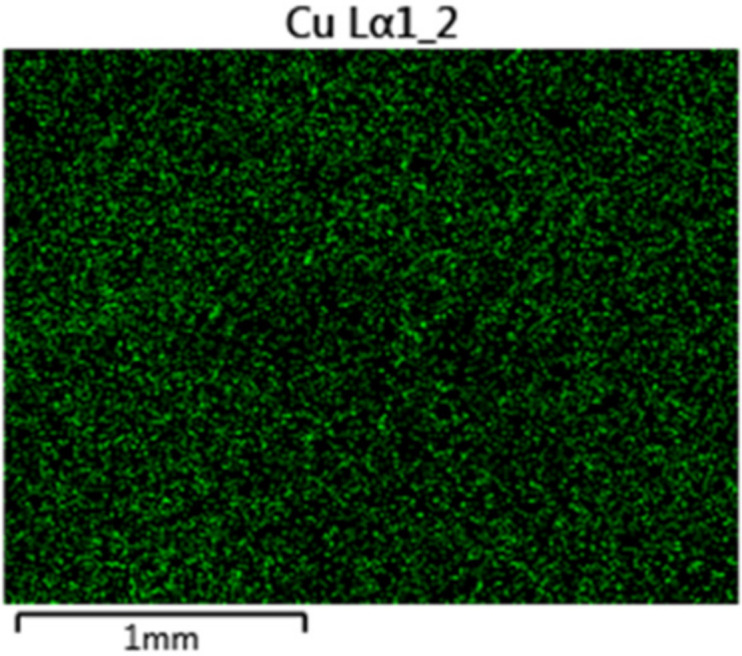
Distribution of Cu via EDX mapping on membrane surface after photo-filtration of solution containing Cu^2+^ and HCO_3_^−^.

**Figure 6 membranes-12-00245-f006:**
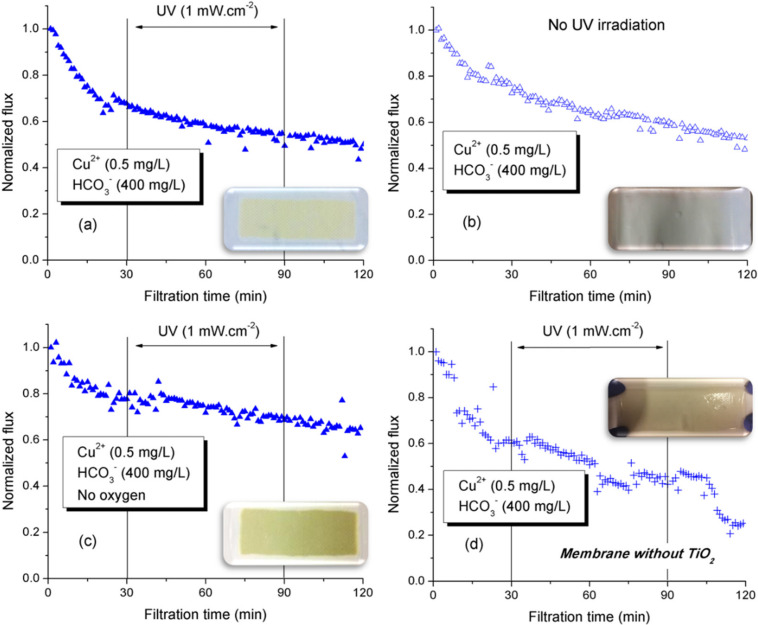
Flux behavior during filtration of solutions containing Cu^2+^ and HCO_3_^−^ at various conditions: (**a**) PVDF-TiO_2_ membrane with photofiltration, (**b**) PVDF-TiO_2_ membrane without photo-filtration, (**c**) PVDF-TiO_2_ membrane in deoxygenated photo-filtration, and (**d**) neat PVDF membrane with photo-filtration.

**Figure 7 membranes-12-00245-f007:**
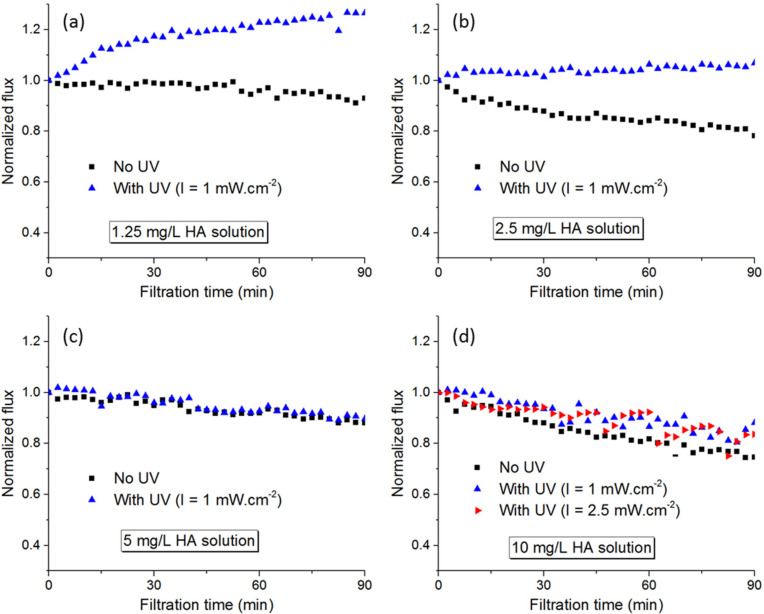
Permeate flux during (photo-)filtration tests of HA at different concentrations: (**a**) 1.25 mg/L, (**b**) 2.5 mg/L, (**c**) 5 mg/L, and (**d**) 10 mg/L.

**Figure 8 membranes-12-00245-f008:**
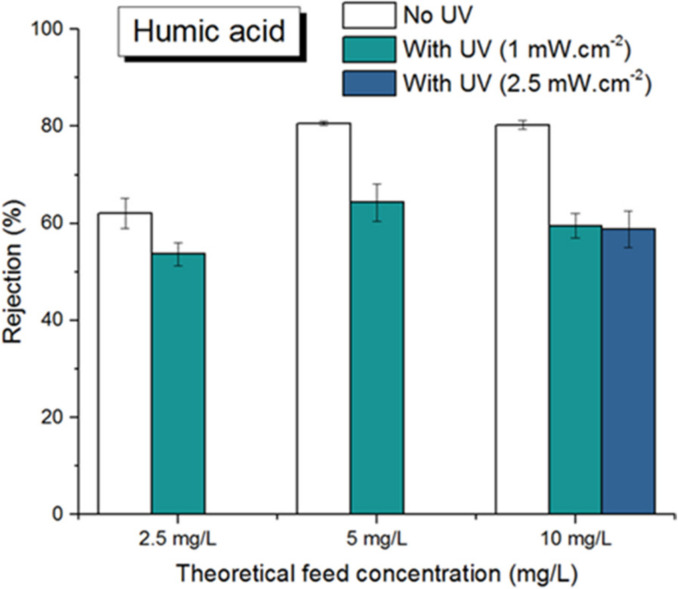
Rejection rate of HA.

**Figure 9 membranes-12-00245-f009:**
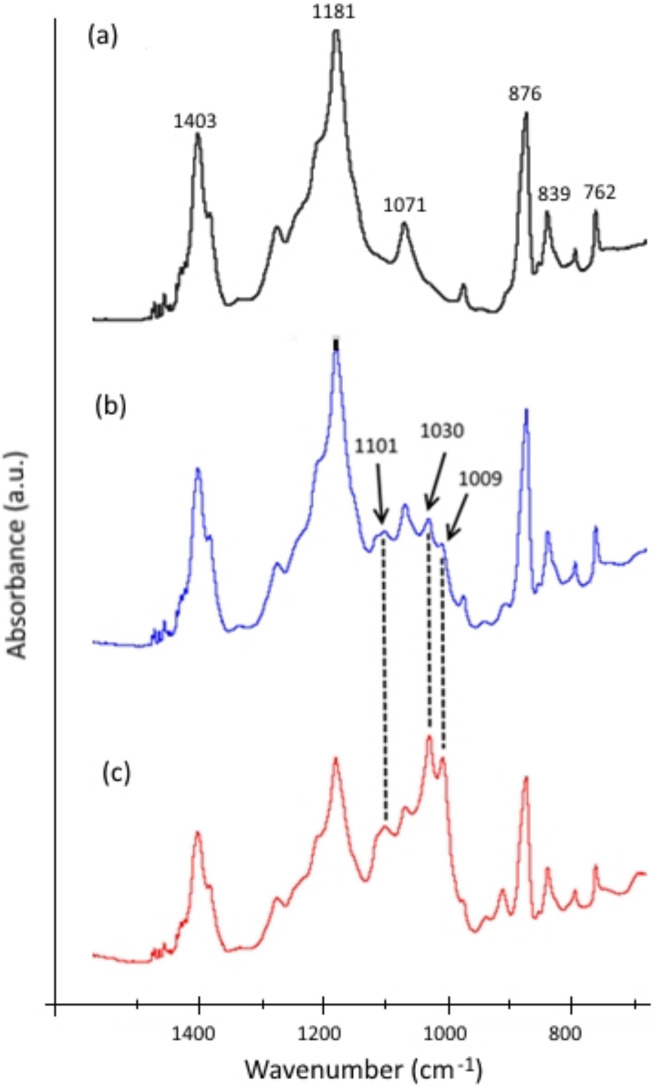
FTIR spectra of PVDF-TiO_2_ membranes: (**a**) pristine, (**b**) after filtration of 10 mg/L HA solution without UV, and (**c**) after photo-filtration of 10 mg/L HA solution.

**Figure 10 membranes-12-00245-f010:**
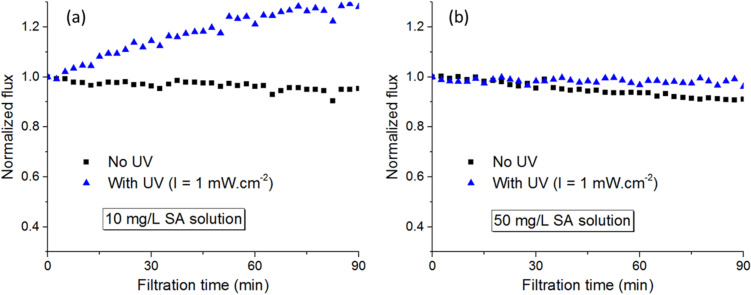
Permeate flux during (photo-)filtration tests of SA at (**a**) 10 mg/L and (**b**) 50 mg/L.

**Figure 11 membranes-12-00245-f011:**
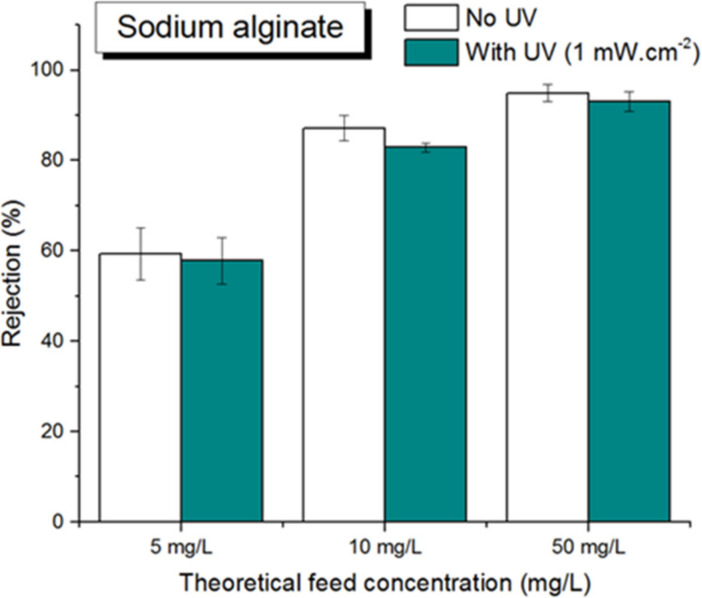
Rejection rate of SA.

**Figure 12 membranes-12-00245-f012:**
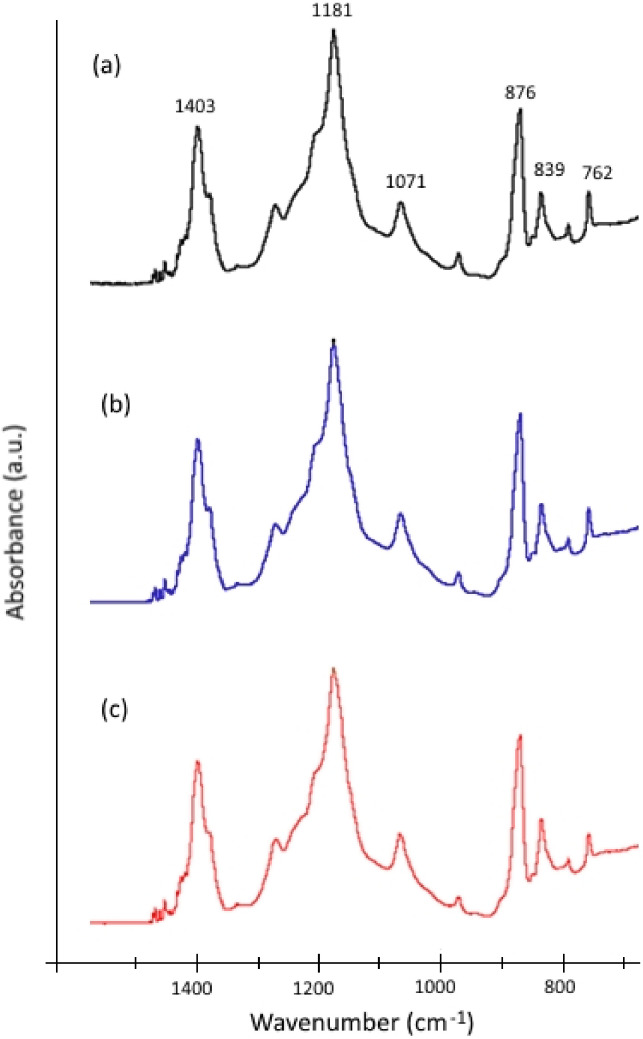
FTIR spectra of PVDF-TiO_2_ membranes: (**a**) pristine, (**b**) after filtration of 50 mg/L SA solution without UV, and (**c**) after photo-filtration of 50 mg/L SA solution.

**Table 1 membranes-12-00245-t001:** Inorganic composition of pure water (Milli-Q), tap water, and mineral water (Hepar^®^).

Species (mg/L)	Pure Water	Tap Water	Mineral Water	Note
HCO_3_^−^	n/d	441.1	383.7	Analyzed by ionic chromatography and chemical titration
Cl^−^	n/d	36.8	18.8	
NO_3_^−^	n/d	3.8	4.3	
SO_4_^2−^	n/d	24.4	1530.0	
Na^+^	n/d	21.1	14.2	
K^+^	n/d	1.4	4.1	
Mg^2+^	n/d	8.6	119.0	
Ca^2+^	0.05	117.1	549.0	
Cu	n/d	0.47	n/d	Analyzed by ICP-MS
Zn	n/d	0.16	n/d	

## Data Availability

The data presented in this study are available on request from the corresponding author.

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
