# Peer review of "Performance of PVDF-TiO2 Membranes during Photo-Filtration in the Presence of Inorganic and Organic Components"

_membranes, 2022, doi:10.3390/membranes12020245_

Round 1

Reviewer 1 Report

In this manuscript, the anti-foiling performance of PVDF-TiO2 composite membrane in the photo-filtration process was investigated. The influence of inorganic and organic components on membrane fouling in the photo-filtration process was studied in detail. This work should be of interest to the readership of Membranes. The authors may wish to consider the following comments in a revised version:

  1. Since the PVDF-TiO2 membrane was self-prepared, some basic information such as membrane structure, surface TiO2 content, pure water flux, MWCO should be presented for a better understanding of the membrane performance variation in the photo-filtration process.
  2. Neat PVDF membrane should also be tested as a comparison.
  3. The stability of the membrane under UV irradiation should be considered. What about the long-term performance of the membrane.

Author Response

Reviewer 1 :

In this manuscript, the anti-foiling performance of PVDF-TiO2 composite membrane in the photo-filtration process was investigated. The influence of inorganic and organic components on membrane fouling in the photo-filtration process was studied in detail. This work should be of interest to the readership of Membranes. The authors may wish to consider the following comments in a revised version:

  1. Since the PVDF-TiO2 membrane was self-prepared, some basic information such as membrane structure, surface TiO2 content, pure water flux, MWCO should be presented for a better understanding of the membrane performance variation in the photo-filtration process.

A: we have added some basic information with regard to membrane properties at the beginning of the results and discussion section. For more details, reader can refer to our previous publications (cited in the text).

  1. Neat PVDF membrane should also be tested as a comparison.

A: similarly, results with neat PVDF membranes were presented in our previous publication for comparison with PVDF-TiO2 membranes. In this publication, the focus is on the photo-filtration performance. The comparison here is rather between PVDF-TiO2 membranes operated with and without UV irradiation. Since neat PVDF membrane does not possess photo-induced properties, we consider it not necessary to include these results.

  1. The stability of the membrane under UV irradiation should be considered. What about the long-term performance of the membrane.

A: the long-term stability and performance of the membrane is the subject of another study with PVDF-TiO2 membranes made in hollow-fiber form, which is the more appropriate configuration for that kind of study.

Reviewer 2 Report

The anti-fouling efficacy of PVDF-TiO2 composite membranes, as measured by permeate flow, was investigated in this work using several synthetic feed solutions. The filtration under continuous UV irradiation of solutions comprising inorganic and organic components found in drinking/natural water was carried out to assess their impact on the photo-induced characteristics and performance of the membranes. The manuscript is well written, therefore I recommend the publication in Membrane after minor technical revision. My comments are available below:

  • Figures 9 and 12 show undefined peaks in the FTIR spectrum that should be specified.
  • Figures 9 and 12 should have horizontal and vertical axes.
  • Why wasn't the HA rejection rate shown in Figure 8 for feed values of 2.5 and 5 mg/L?
  • Each induvial plot, such as a, b, and c, should be described in the figure caption.
  • The filtration mechanism using PVDF-TiO2 membranes should be shown schematically and described.
  • In figure 2, why the normalized flux is greater than 1?

Author Response

The anti-fouling efficacy of PVDF-TiO2 composite membranes, as measured by permeate flow, was investigated in this work using several synthetic feed solutions. The filtration under continuous UV irradiation of solutions comprising inorganic and organic components found in drinking/natural water was carried out to assess their impact on the photo-induced characteristics and performance of the membranes. The manuscript is well written, therefore I recommend the publication in Membrane after minor technical revision. My comments are available below:

  • Figures 9 and 12 show undefined peaks in the FTIR spectrum that should be specified.
  • Figures 9 and 12 should have horizontal and vertical axes.

A: these details are corrected in the revision.

  • Why wasn't the HA rejection rate shown in Figure 8 for feed values of 2.5 and 5 mg/L?

A: we understand that the reviewer is referring to the HA rejection rate at UV intensity of 2.5 mW.cm-2. However, the standard UV intensity being used in this study is 1 mW.cm-2, a value we have proved that is more than enough for the photo-induced properties of the membranes to be activated (Chem. Eng. Process. - Process Intensif. 148 (2020) 107781). That is why only one test at 2.5 mW.cm-2 was performed in this study, to show that further increase of UV intensity did not lead to better performance.

  • Each induvial plot, such as a, b, and c, should be described in the figure caption.

A: these details are corrected in the revision.

  • The filtration mechanism using PVDF-TiO2 membranes should be shown schematically and described.

A: the filtration mechanism of PVDF-TiO2 membranes is the same with other composite MF/UF membranes which were already well-described in literature, so we consider it not necessary to be repeated. What sets the PVDF-TiO2 membrane apart from other composite membranes is its photo-induced properties during photo-filtration, which is the main problem discussed in this article and our other previous publications.

  • In figure 2, why the normalized flux is greater than 1?

A: flux during photo-filtration being higher than the initial flux (when UV is not activated) is actually an interesting feature of PVDF-TiO2 membranes, thanks to the photo-induced superhydrophilicity effect. This feature was already investigated in details in one of our previous publication (Chem. Eng. Process. - Process Intensif. 148 (2020) 107781).
